# Collaborative Inter-agent Knowledge Distillation for Reinforcement Learning

## Abstract

Reinforcement Learning (RL) has demonstrated promising results across several sequential decision-making tasks. However, reinforcement learning struggles to learn efficiently, thus limiting its pervasive application to several challenging problems. A typical RL agent learns solely from its own trial-and-error experiences, requiring many experiences to learn a successful policy. To alleviate this problem, we propose *collaborative inter-agent knowledge distillation* (CIKD). CIKD is a learning framework that uses an ensemble of RL agents to execute different policies in the environment while sharing knowledge amongst agents in the ensemble. Our experiments demonstrate that CIKD improves upon state-of-the-art RL methods in sample efficiency and performance on several challenging MuJoCo benchmark tasks. Additionally, we present an in-depth investigation on how CIKD leads to performance improvements.

## 1 Introduction

Reinforcement learning (RL) [47] has demonstrated impressive performance on solving sequential decision-making tasks in interactive environments (e.g., video game-playing [32] or robotic control [18]). In these tasks, the RL agent's goal is to find the optimal policy, which maximizes the expected return in the task. The agent learns this optimal policy through many trial-and-error interactions with the environment. Through these trial-and-error interactions, the agent explores the consequences of different decisions, in the form of a reward or punishment. Unfortunately, millions of interactions are required even to solve simple tasks. Such amounts of interaction are infeasible to acquire in real, physical tasks, as is the case for robots.

In order to learn the optimal policy, the agent performs policy improvement, which refers to the incremental process of refining the agent's policy towards performing high-rewarding actions. However, if the agent has a poor policy, then this incremental updating can be a slow process, requiring a large number of interactions before the agent acquires the experiences necessary for learning desirable behaviors. This is because a typical RL agent gathers experiences using a policy close to its greedy policy, and if this greedy policy itself is poor, acquiring high-quality experiences can require a large number of exploratory actions. However, through collaboration amongst multiple agents with different behaviors, perhaps we can reduce the amount of experiences needed for an agent to acquire a good policy.

For inspiration, we turn to the study of collective animal behavior [46]. Rather than exclusively learn from trial-and-error alone, the optimal behaviors for several tasks (e.g., flocking and foraging) can emerge through collaboration amongst animals. For example, consider ants foraging in a colony [8]. The ants search divergent paths, resulting in extensive exploration as a group, across a wide range of food-gathering policies. Furthermore, the ants that find food share their path with their companions via pheromones. This information-sharing amongst colony members offers each ant beneficial guidance toward food, so that individual ants need not exhaustively search several paths in order to find food. To make an analogy to the RL setting, the ants' foraging process can be viewed as an effective way of policy improvement. Motivated by these insights from collective animal behavior, we incorporate the notion of collective knowledge sharing into the RL setting in an attempt to accelerate and guide the search for the optimal policy.

Our method emulates the collaborative behaviors of ants via an ensemble of RL agents: a group of agents collectively search for the optimal policy in the same task, while periodically sharing knowledge. Each RL agent resembles an individual in the ant colony. To elicit diverse experiences amongst members of the ensemble, each RL is randomly initialized with different neural network parameters. Random initialization results in adequate behavioral diversity [22, 34] needed for collective exploration. As these agents are diverse in nature, at any given time during the course of training, one agent naturally has a policy superior to its peers. This agent can guide its peers towards

higher-performing policies, just as ants share superior paths through pheromones. In order to guide the other agents, our method employs a knowledge distillation framework [24] which is effective at transferring knowledge between neural networks, without assuming identical model architectures. Knowledge distillation encourages other agents to act in a manner similar to the leading agent, allowing agents to escape underperforming policies. Moreover, knowledge distillation empowers all agents to continue searching for the optimal policy from better starting points, accelerating knowledge accumulation in the entire ensemble. However, despite the distillation, the agents' knowledge is still preserved, retaining diversity amongst the ensemble.

This paper's primary contribution is *collaborative inter-agent knowledge distillation* (CIKD), a simple yet effective framework for RL that jointly trains an ensemble of RL agents while periodically performing knowledge sharing. We demonstrate empirically that our method can improve the state-of-the-art soft-actor critic (SAC) [20] on a suite of challenging MuJoCo tasks, exhibiting superior sample efficiency and performance. We further validate the effectiveness of distillation for knowledge sharing by comparing against other methods of sharing knowledge. In addition, we present an in-depth investigation to explain the underlying causes of CIKD's performance improvement. Finally, our ablation study shows that a small ensemble is sufficient for improving performance.

The remainder of this paper is organized as follows. Section 2 discusses the related work. Section 3 introduces the reinforcement learning formulation. Section 4 describes our method. Section 5 presents our experimental findings. Section 6 summarizes our contributions and outlines potential avenues for future work.

## 2 RELATED WORK

The idea of jointly training multiple policies through RL has emerged in prior works. The most relevant works [37, 38] train multiple policies for the same task through RL, as our method does. Osband et al. [37, 38] train several agents in an ensemble while storing these agents' experiences in a shared buffer. Thus, agents share knowledge by sharing experiences amongst members of the ensemble, which are then used for RL updates. Our method is complementary to Osband et al. [37, 38]'s, in that we can also use a shared buffer of experiences, but we additionally periodically performing knowledge distillation between members of the ensemble, in particular from the best agent to other agents. In our experiments we present supporting evidence to justify the importance of knowledge sharing. Other related methods aggregate multiple policies to select actions [2, 6, 10, 17, 31, 40, 41, 45, 50, 54]. Abel et al. [1], Tosatto et al. [52], Wang & Jin [53] sequentially train a series of policies, boosting the learning performance by using the errors of a prior policy. However, rather than perform decision aggregation or sequentially-boosted training, we focus on improving the performance of each individual agent via knowledge sharing amongst jointly trained agents. Again, our method can be considered to be a complementary approach to decision aggregation and sequentially boosted training methods.

We can view the sharing of knowledge from the best policy as exploiting successful behavior patterns. This general notion has been explored in several areas of RL. Rusu et al. [42] and Parisotto et al. [39] can train a single network to perform multiple tasks by transferring multiple pre-trained RL agents' policies to a single network. Hester et al. [23], Nair et al. [35] accelerate the RL agents' training progress via human experts' guidance. Instead of using experts' policies, Levine & Koltun [30], Nagabandi et al. [33] and Zhang et al. [55] leverage model-based controllers' behaviors (e.g. model predictive controllers [15] or linear-quadratic regulators Dorato et al. [7]), facilitating training for RL agents. Additionally, Hong et al. [25] and Oh et al. [36] train agents to imitate the past successful self-experiences or policies. Orthogonal to the aforementioned works, our method periodically exploits the current best policy amongst the ensemble, and shares it with the ensemble, enabling a collaborative search for the optimal policy.

Collaborative learning approaches have emerged in other areas of machine learning research. In computer vision, Zhang et al. [56] present deep mutual learning, which trains multiple models that mutually imitate each other's outputs on classification tasks. Our distillation is not mutual, and flows in a single direction, from a superior teacher agent to other student agents in the ensemble. In subsequent work by Lan et al. [29], they train an ensemble of models to imitate a stronger teacher model. Simultaneously, this teacher model learns to aggregate all of the ensemble models' predictions. They demonstrate that imitating a superior teacher leads to better performance than deep mutual learning, which performs mutual imitation amongst members. Our method contrasts from the above methods by periodically electing the teacher for distillation. Since our distillation occurs periodically,

we can collect performance statistics for each agent between periods, which can be used to elect the teacher in the distillation phase. In contrast to the distillation of an aggregate policy, as Lan et al. [29] do, our method ensures that the teacher indeed possesses a superior policy.

Collaborative learning has also been explored in RL as well. Teh et al. [49] and Ghosh et al. [16] jointly learn independent policies for multiple tasks or contexts and then distill these policies to a central multi-task policy. Galashov et al. [12] learn a task-specific policy while bounding the divergence between this task-specific policy and some generic policy that can perform basic task-agnostic behaviors. Czarnecki et al. [5] gradually transfer the knowledge of a simple policy to a complex policy during the course of joint training. While our method also collaboratively trains policies, we differ from the aforementioned works in several aspects. First, our method periodically elects a leading agent for sharing knowledge rather than either constraining the mutual policy divergence [14, 16, 49, 56] or imitating aggregated models [29]. The second difference is that our method does not rely on training heterogeneous policies (e.g. a simple policy and a complex policy [5]), which makes our method more generally applicable. Finally, as opposed to Teh et al. [49] and Ghosh et al. [16], we consider single-task setting rather than multi-tasking.

Evolutionary algorithms (EA) [13, 21, 26, 43] similarly employ multiple policies to find the optimal policy. EA repeatedly performs mutation, selection, and reproduction on a maintained population. Mutation randomly perturbs the parameters of policies in the population; selection eliminates the underperforming policies by testing the policies' performance in the environment; reproduction produces the next generation of policies from the remaining policies. Unlike EA, our method does not rely on mutation and reproduction on a population of policies. While EA and our method are similar in an abstract sense, in that they both share knowledge amongst agents, they are quite different in practice. EA often eliminates members from its population and performs destructive changes to members of the population. Our method focuses on continuously improving the existing agents, and does not perform destructive changes to its population. In fact, our method can be incorporated into EA, serving as a more effective way to optimize each individual policy within the same generation.

## 3 BACKGROUND

In this section we describe the general framework of RL. RL formalizes a sequential decision-making task as a *Markov decision process* (MDP) [47]. An MDP consists of a state space $\mathcal{S}$, a set of actions $\mathcal{A}$, a (potentially stochastic) transition function $\mathcal{T} : \mathcal{S} \times \mathcal{A} \to \mathcal{S}$, a reward function $\mathcal{R} : \mathcal{S} \times \mathcal{A} \to \mathbb{R}$, and a discount factor $\gamma \in [0, 1]$. An RL agent performs *episodes* of a task where an agent starts in a random initial state $s_0$, sampled from the initial state distribution $\rho_{s_0}$, and performs actions, experiencing new states and rewards. More generally, at timestep $t$, an agent performs an action $a_t$ in state $s_t$, receives a reward $r_{t+1}$, and transitions to a new state $s_{t+1}$, according to the transition function $\mathcal{T}$. The discount factor $\gamma$ is used to indicate the agent's preference for short-term rewards over long-term rewards.

An RL agent performs actions according to its policy, a conditional probability distribution $\pi_\phi : \mathcal{S} \times \mathcal{A} \mapsto [0, 1]$, where $\phi$ denotes the parameters of the policy, which may be the parameters of a neural network. RL methods iteratively update $\phi$ via rollouts of experience $\tau = \{(s_t, a_t, r_t, s_{t+1})\}_{t=0}^T$, seeking within the parameter space $\Phi$ for the optimal $\phi^*$ that maximizes the expected return $\mathbb{E}_{s \sim \rho_{s_0}} \left[ \sum_{t=0}^T \gamma^t r_t | s_0 = s \right]$ for each $t$ within an episode.

## 4 METHOD

In this section, we formally present the technical details of our method, Collaborative Inter-agent Knowledge Distillation (CIKD). We start by providing an overview of CIKD and then its components in detail.

### 4.1 OVERVIEW

Emulating collaborative foraging behaviors of ants, CIKD employs an ensemble of RL agents to perform a wide range of policies on the same task and periodically share knowledge amongst agents. CIKD can be divided into three phases: ensemble initialization, joint training, and inter-agent knowledge distillation. First, in the ensemble initialization phase, we randomly initialize an ensemble of RL agents to achieve behavioral diversity. In the joint training stage, each agent acts in the environment, adding its experiences to a shared buffer, and these shared experiences are used to update the parameters of the agent. Intermittently, we perform inter-agent knowledge distillation, where we elect a leading agent to guide the other agents towards superior policies. To this end, we

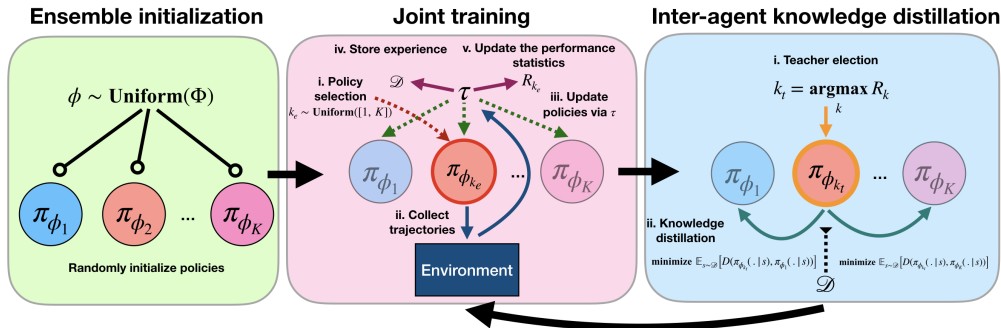

Figure 1: An overview of collaborative inter-agent knowledge distillation.

distill [24] the best-performing agent's policy to the others. Algorithm 1 and Figure. 1 summarizes our method.

---

**Algorithm 1** Collaborative Inter-agent Knowledge Distillation for Reinforcement Learning
---

1: **Input:** an environment $\mathcal{E}$, an ensemble size $K$, a parameter space $\Phi$, a set of parameterized policies $\{\pi_{\phi_k}\}_{k=0}^{K}$, recent episodic performance statistics $\{R_k\}_{k=0}^{K}$, an episode length $T$, a joint training length $E$, a distillation interval $I$, an experience buffer $\mathcal{D}$
2: **Output:** the optimal policy $\pi^*$
3:
4: **i. Ensemble initialization**
5: Randomly initialize policy parameters: $\phi_k \sim \text{Uniform}(\Phi), \forall k \in [0, K)$
6: Initialize the experience buffer: $\mathcal{D} \leftarrow \varnothing$
7: Initialize each recent episodic performance statistic $R_k, \forall k \in [0, K)$
8: Initialize timestep counter: $t_{acc} \leftarrow 0$
9: **while** not converged **do**
10:     **ii. Joint training**
11:     **for** episode $\in [0, E)$ **do**
12:         Policy selection: $k_e \sim \text{Uniform}([0, K))$
13:         Perform a rollout in the environment: $\tau \leftarrow \text{ROLLOUT}(\mathcal{E}, \pi_{\phi_{k_e}})$
14:         Store the experience: $\mathcal{D} \leftarrow \mathcal{D} \cup \tau$ $(\tau = \{(s_t, a_t, r_t, s_{t+1})\}_{t=0}^{T})$
15:         Update all policies $\pi_{\phi_k}, \forall k \in [0, K)$ by off-policy RL methods using batches sampled from $\mathcal{D}$
16:         Update the selected recent-performance statistics: $\text{UPDATESTAT}(R_{k_e}, \tau)$
17:         Accumulate timestep counter: $t_{acc} \leftarrow t_{acc} + T$
18:     **end for**
19:     **iii. Inter-agent Knowledge Distillation**
20:     **if** $t_{acc} \mod I = 0$ **then**
21:         Elect the teacher agent: $k_t = \text{argmax}_k R_k$
22:         Minimize $L(\phi_k, \phi_{k_t}), \forall k \in [0, K) \setminus k_t$ using $\mathcal{D}$ (Equation. 1)
23:     **end if**
24: **end while**

---

## 4.2 ENSEMBLE INITIALIZATION

We randomly initialize $K$ RL agents in the ensemble. Each RL agent's policy is instantiated with a model parameterized by $\phi_k$, where $k$ stands for the agent's index in the ensemble. $\phi_k$ is initialized by sampling from the uniform distribution over parameter space $\Phi$ which contains all possible values of $\phi_k$: $\phi_k \sim \text{Uniform}(\Phi)$. Despite the simplicity of uniform distributions, Osband et al. [37] show that uniformly random initialization can provide adequate behavioral diversity. In this paper, we represent each $\phi_k, \forall k \in [0, K)$ as a neural network (NN) due to the impressive performance of NNs in recent RL research [20, 32], though other parametric models (e.g. linear models) can be used.

CIKD can be easily applied to off-policy actor-critic methods, which learn both a policy and a critic function that values states or state-action pairs. Off-policy actor-critic methods store a replay buffer of past experiences that are used for training. In this paper, we use soft actor-critic (SAC) [20], an off-policy actor-critic method that has demonstrated state-of-the-art results on challenging tasks [19]. To apply CIKD to SAC, we create a shared replay buffer for all agents and randomly initialize a critic function for each policy $\pi_{\phi_k}$.

### 4.3 JOINT TRAINING

Each joint training phase consists of $E$ episodes. For each episode, we select an agent in the ensemble to act in the environment (hereinafter, we refer this process as "policy selection") and then update all agents' policies using the experiences from that episode. Below, we describe the policy selection and policy update procedures.

The policy selection strategy is a way to select one policy $\pi_{\phi_{k_e}}$ from the ensemble, to perform one episode $\tau$ in the environment. This episode $\tau$ is then stored in a shared experience buffer $\mathcal{D}$, and the agent's recent episodic performance statistic $R_{k_e}$ is updated according to the return achieved in $\tau$, where $R_{k_e}$ stores the average episodic return in the most recent $M$ episodes. $\{R_k\}_{k=0}^K$ and $\mathcal{D}$ will later be used in inter-agent distillation (Section 4.4). There are two purposes of policy selection. The first purpose is to collect diverse experiences from different policies. In this paper, we adopt a simple uniform policy selection strategy, whereby at the beginning of each episode we select a policy from the ensemble at random to act in the episode. After selecting a policy $\pi_{\phi_{k_e}}$ which performs an episode $\tau$, we store this $\tau$ in $\mathcal{D}$. Then, we can sample data from $\mathcal{D}$ and update all policies using any arbitrary off-policy update method. Since off-policy update methods do not require that $\tau$ necessarily comes from a specific policy, they enable our policies to learn from the trajectories generated by other members of the ensemble.

### 4.4 INTER-AGENT KNOWLEDGE DISTILLATION

The inter-agent knowledge distillation phase consists of two stages: *teacher election* and *knowledge distillation*. The purpose of teacher election is to determine which agent in the ensemble has the best-performing policy, which can then serve as a teacher to the other ensemble members. In our experiments, teacher election is determined from each agent's recent episodic performance statistics recorded during the joint training stage. Thus, the selected teacher agent is the agent with the highest recent performance, namely $k_t = \arg\max_k R_k$, where $k_t$ is the index of the teacher. Rather than use the agent's most recent episodic performance, we use its average return over its previous $M$ episodes, to minimize the noise in our estimate of the agent's performance. We additionally considered performing offline evaluations, but this requires additional environmental interactions.

Next, the teacher serves as a guide that leads the other agents towards higher-performing policies. This is done through knowledge distillation [24], which has been shown to be effective at guiding the neural network to behave similarly to another neural network. To distill from the teacher to the students (i.e., the other ensemble members), the teacher samples experiences from the buffer $\mathcal{D}$ and instructs each student to match its outputs on these samples. The intuition is that after distillation, the students acquire the teacher's knowledge, enabling them to correct their sub-optimal behaviors and reinforce their correct behaviors. Specifically, the distillation process can be formalized as minimizing the following loss function:

$$L(\phi_k, \phi_{k_t}) = \mathbb{E}_{s \sim \mathcal{D}} \left[ D(\pi_{\phi_{k_t}}(.|s), \pi_{\phi_k}(.|s)) \right], \tag{1}$$

where $D$ can be any distance metric between two functions. In this paper, $D$ is taken as Kullback-Leibler divergence ($D_{KL}$) to measure the similarity of policies. We use $s$ to denote historical states sampled from the experience buffer $\mathcal{D}$. $D_{KL}$ is a principled way to measure the similarity between two probability distributions (i.e., policies). As $D_{KL}$ is an asymmetric metric, we clarify the settings below. $\pi_{\phi_{k_t}}$ and $\pi_{\phi_k}$ are taken as the ideal distribution and approximated distribution, respectively (i.e., $P$ and $Q$ in the standard notation of KullbackLeibler divergence [28]). Note that inter-agent distillation is also compatible with actor-critic methods. For actor-critic methods, we additionally distill the critic function from the teacher to the students. In this paper, our distance metric for the critics is the $l_2$-loss function. In practice, the distance metric between critics can be any other metric for regression (i.e., critic training can be viewed as a regression problem).

## 5 EXPERIMENTS

The experiments are designed to answer the following questions: (1) Can CIKD improve upon the data efficiency of state-of-the-art RL? (2) Is knowledge distillation effective at sharing knowledge? (3) Is is it necessary to choose the best-performing agent to be the teacher? (4) Why does CIKD improve the performance of RL methods? (5) What is the impact of the ensemble size? Next, we show our experimental findings for each of the aforementioned questions, and discuss their implications.

## 5.1 EXPERIMENTAL SETUP

**Implementation.** Our goal is to demonstrate how CIKD improves the sample efficiency of an RL algorithm. Since soft actor-critic (SAC) [20] exhibits state-of-the-art sample efficiency across several simulated benchmarks [51] and even in real robots [18, 19], we build on top of SAC. We directly use the hyperparameters for SAC from the original paper [20] in all of our experiments. Unless stated otherwise, the hyperparameters used in our proposed method (Algorithm 1) are $I = 5000$, $M = 5$, and $K = 3$ for all experiments. The value of $I$ and $M$ are tuned via grid search over $[1000, 2000, \cdots, 10000]$, and $[1, 2, \cdots, 10]$ respectively. We experiment with $K \in \{2, 3, 5\}$ in Section 5.6. For the remainder of our experiments, we term CIKD applied to SAC as *SAC-CIKD*.

**Benchmarks.** We use OpenAI gym [3]'s MuJoCo [51] benchmark tasks, as used in the original SAC [20] paper. We choose all tasks selected in the original paper [20] and two additional tasks to evaluate the performance of our method. The description for each task can be found in the source code for OpenAI gym[1].

**Evaluation.** We adapt the evaluation approach from the original SAC paper [20]. We train each agent for 1 million timesteps, and run 20 evaluation episodes after every 10000 timesteps (i.e., number of interactions with the environment), where the performance is the mean of these 20 evaluation episodes. We repeat this entire process across 5 different runs, each with different random seeds. We plot the mean value and confidence interval of mean episodic return at each stage of training. The mean value and confidence interval are depicted by the solid line and shaded area, respectively. The confidence interval is estimated by the bootstrapped method [9]. At each evaluation point, we report the highest mean episodic return amongst the agents in the ensemble. In some curves, we additionally report the lowest mean episodic return amongst the agents in the ensemble.

## 5.2 EFFECTIVENESS OF INTER-AGENT KNOWLEDGE DISTILLATION

In order to evaluate the effectiveness of inter-agent knowledge distillation, we compare our method, *SAC-CIKD*, with two baselines: *Vanilla-SAC* and *Ensemble-SAC*. *Vanilla-SAC* stands for the original SAC; *Ensemble-SAC* is the analogous variant of Osband et al. [37]'s method on SAC. At its core, Osband's method involves an ensemble of agents that act in the environment and generate trajectories. These trajectories are then used to train all of the agents using an off-policy RL algorithm. Our *Ensemble-SAC* baseline resembles this by having each agent represented as an actor-critic neural network trained with the off-policy algorithm SAC (effectively *SAC-CIKD* without inter-agent knowledge distillation). Ensemble sizes ($K$) for *Ensemble-SAC* and our method are set to 3. Our results are shown in Figure 2. Note that we also plot the worst evaluation in our ensemble at each evaluation phase to provide some insight into the general performance of the ensemble. In all tasks, we outperform all baselines, including *Vanilla-SAC* and *Ensemble-SAC*. Moreover, *SAC-CIKD* has far better sample efficiency, usually reaching the best baseline's convergent performance in half of the environment interactions. We even find that in the majority of tasks, our worst evaluation in the ensemble outperforms the baseline methods. This demonstrates that all members of the ensemble are significantly improving, and our method's superior performance is not simply a consequence of selecting the best agent in the ensemble. Notice that *Ensemble-SAC* does not significantly improve the performance of *Vanilla-SAC*. *Ensemble-SAC* only outperforms *Vanilla-SAC* in 4 out of 7 tasks and suggests that the diversity of the ensemble is alone insufficient for achieving large performance gains over *Vanilla-SAC*. In particular, *SAC-CIKD*'s superiority over *Ensemble-SAC* highlights the effectiveness of supplementing shared experiences (*Ensemble-SAC*) with knowledge distillation. In summary, Figure 2 demonstrates the effectiveness of inter-agent knowledge distillation on enhancing the performance and data efficiency of state-of-the-art RL algorithms.

## 5.3 EFFECTIVENESS OF KNOWLEDGE DISTILLATION FOR KNOWLEDGE SHARING

In this section, we investigate the advantage of using knowledge distillation for knowledge sharing. Beyond using knowledge distillation [24], we consider other approaches that may successfully share knowledge amongst agents. First, we consider sharing knowledge by simply providing agents with additional policy updates using the shared experiences. We also consider directly copying the neural network. Below, we separately compare these two approaches with knowledge distillation.

Though Section 5.2 has shown that *Ensemble-SAC* which updates all agents' policies through shared experiences fails to perform as well as *SAC-CIKD*, *SAC-CIKD* uses additional gradient updates during

---

[1]https://github.com/openai/gym

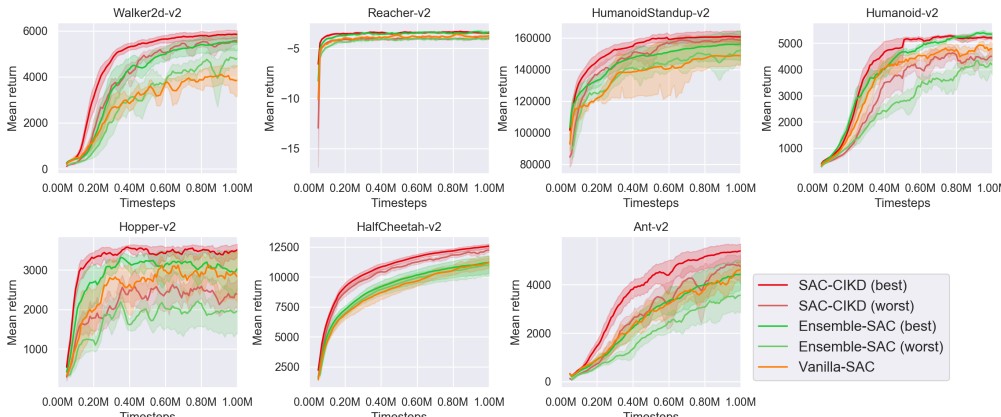

Figure 2: **Performance evaluation of inter-agent knowledge distillation**. *SAC-CIKD* represents the implementation of our method upon SAC; *Vanilla-SAC* stands for the original SAC; *Ensemble-SAC* is an analogous variant of Osband et al. [37]'s method on *vanilla-SAC* (effectively *SAC-CIKD* without inter-agent knowledge distillation). See Section 5.2 for details. Notice that in most domains, *SAC-CIKD* is able to reach the convergent performance of the baselines in less than half the training time.

the inter-agent knowledge distillation phase, whereas *Ensemble-SAC* only performs joint training. It is unclear whether extra policy updates in lieu of knowledge distillation can achieve the same effects as knowledge distillation. To investigate this, we compare our method with *Vanilla-SAC (extra)* and *Ensemble-SAC (extra)*, which respectively correspond to *Vanilla-SAC* and *Ensemble-SAC* (see Section 5.2) agents that are trained with extra policy update steps. Specifically, instead of the knowledge distillation phase, we provide these baseline agents with the same number of policy updates and minibatch sizes as we give the *SAC-CIKD* agent for knowledge distillation. A policy update here refers to as a training step for updating the policy [44, 48] and a value function [20, 27], if required, by RL algorithms. Figure (3a) shows the performance of the above comparison baselines and *SAC-CIKD*. It can be seen that *SAC-CIKD* outperforms all the baselines. This observation shows that knowledge distillation is more effective than policy updates for knowledge sharing.

We additionally study whether the naive method of directly copying parameters from the best-performing agent can also be an effective way to share knowledge between neural networks. We compare a variant of our method, which we denote as *SAC-CIKD (hardcopy)*, against *SAC-CIKD*. In *SAC-CIKD (hardcopy)*, rather than perform inter-agent knowledge distillation, we simply copy the parameters of the teacher policy and critic into the student policies and critics. Figure (3b) depicts the performance of this variant, the baselines, and *SAC-CIKD*. We can see that despite the fact that *SAC-CIKD (hardcopy)* can surpass *Vanilla-SAC*, *SAC-CIKD (hardcopy)* loses to *SAC-CIKD* in all tasks. Thus, knowledge distillation is superior to naively copying the best agent's parameters. While sharing can be beneficial, being too biased towards the teacher can be counterproductive. Identical behaviors may limit the scope of exploration on policies, undermining their quality.

### 5.4 EFFECTIVENESS OF SELECTING THE BEST-PERFORMING AGENT AS THE TEACHER

In the inter-agent knowledge distillation stage, we elect the best-performing agent to be the teacher, and distill this teacher's knowledge to the students (see Section 4.4).

However, the importance of choosing the best-performing agent remains unclear. Perhaps simply sharing knowledge amongst members is sufficient for good performance, and strictly distilling from a high-performing teacher may be unneeded. To investigate the importance of selecting the best-performing agent to be the teacher, we experiment with selecting a random teacher. We denote this variant as *SAC-CIKD (random teacher)*. Figure 4 plots the performance of our method, the baselines, and this variant. It can be seen that *SAC-CIKD (random teacher)* performs on-par with *Ensemble-SAC* 3 out of 4 tasks, and better on a single task. *SAC-CIKD (random teacher)* performs worse than *SAC-CIKD* on 3 tasks, and on-par on a single task. This result suggests that simply sharing knowledge amongst agents can slightly improve performance. Perhaps the reason that random knowledge-sharing can improve performance is because minimizing the KL divergence between two policies (even a randomly chosen policy) draws them closer to one another. Doing so then reduces the extrapolation error [11] caused by training the policies with data distributions generated from other policies, thus improving the quality of off-policy updates within the ensemble. Nevertheless, it is still preferable to select the best-performing agent when sharing knowledge through distillation.

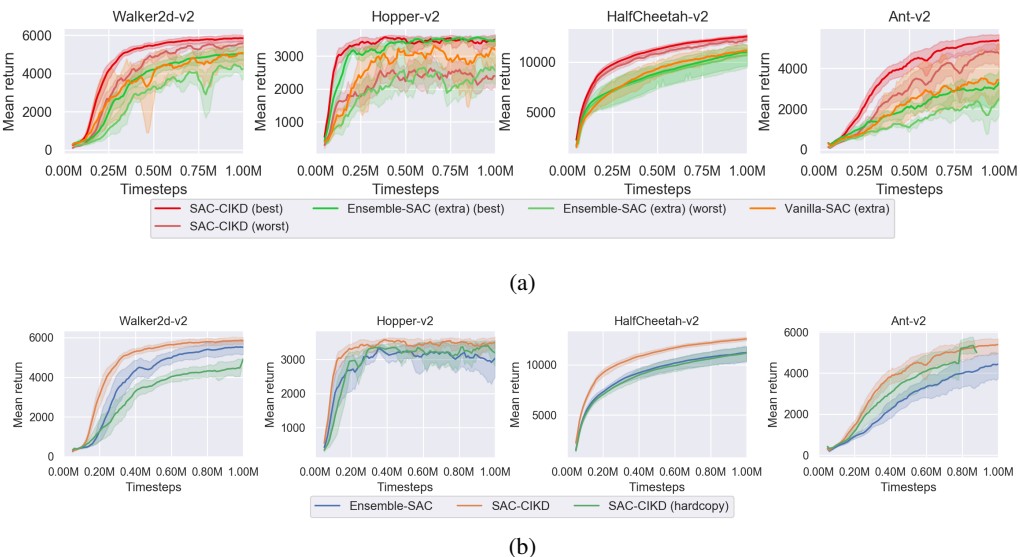

(a)

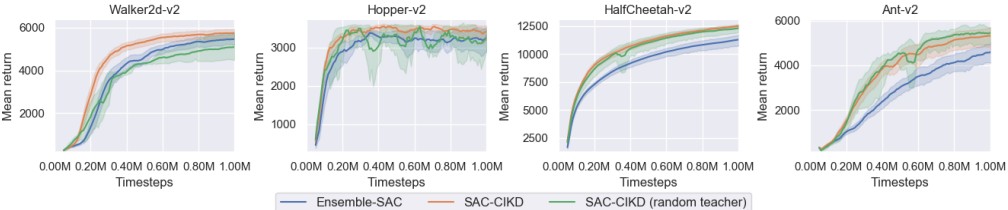

(b)

Figure 3: (a) **Comparison between knowledge distillation and extra policy updates.** *Vanilla-SAC (extra)* and *Ensemble-SAC (extra)* stand for *Vanilla-SAC* and *Ensemble-SAC* variants that use extra policy updates, respectively (see Section 5.3 and Section 3 for details). (b) **Comparison between knowledge distillation and copying parameters.** *SAC-CIKD (hardcopy)* stands for the variant of our method which directly copy the neural networks parameters of the best agent to the others. (a)(b) Both figures show that knowledge distillation is more effective.

Figure 4: **Comparison between the selecting the best-performing teacher vs. a random teacher.** *SAC-CIKD (random teacher)* refers to the variant of our *SAC-CIKD* where a randomly chosen teacher is used for knowledge distillation. This figure demonstrates that it can be more effective to select the best-performing agent as the teacher.

## 5.5 CAUSES FOR THE PERFORMANCE IMPROVEMENT

In this section, we investigate how inter-agent knowledge distillation improves the performance of each agent in the ensemble. Below, we verify each of our hypotheses in the *HalfCheetah-v2* environment. One hypothesis is that knowledge distillation transfers the behaviors of a dominant agent to weaker agents. Such a skilled agent may emerge simply from the diversity stemming from random initialization. To investigate this hypothesis, we plot the selected teacher index (i.e. $k_t$) over the entire training process in Figure 5a. We find that the selected teacher is roughly evenly distributed across all agents. These proportions are presented in Figure. 5b. It can be observed that there is no strictly dominant agent within the ensemble. Both figures suggest that there is no single dominant agent passing its knowledge to inferior agents. Since we find that there is no single dominant agent. Another hypothesis is that knowledge is accumulated throughout members of the ensemble, with agents surpassing their teachers to then become the next teacher, passing on its knowledge in the next period. As such, we are interested in knowing how often agents surpass their teachers immediately subsequent to distillation. To measure this, we evaluate (see Section 5.1 for detail) each agent immediately before and after the inter-agent knowledge distillation phase. Figure 5c depicts the percentage of agents that obtain superior performance to their teachers during different stages of training. We see that throughout the whole training process, in at 23% of post-distillation evaluations, the student agent surpasses its teacher. Furthermore, in the later stages of training, 33% of distillations result in the student agents outperforming their teachers. From these observations, we conclude that stronger agents likely emerge through continual knowledge distillation.

Since knowledge distillation [24] theoretically aims to match the performance between the students and teachers, a performance increase in the students is counter-intuitive. One possible explanation

is that the combination of students' inherent skills and teachers' knowledge results in performance improvement. To test this, we randomly reset the parameters of each student agent prior to distillation, so as to observe the influence of the students' inherent knowledge. Figure 5d plots the performance of this variant and our method. The variant with parameter reset is denoted as *SAC-CIKD (reset)*. It can be observed that *SAC-CIKD (reset)* is much worse than both *SAC-CIKD* and *Ensemble-SAC*. This demonstrates the importance of performing knowledge distillation on an agent that has already learned.

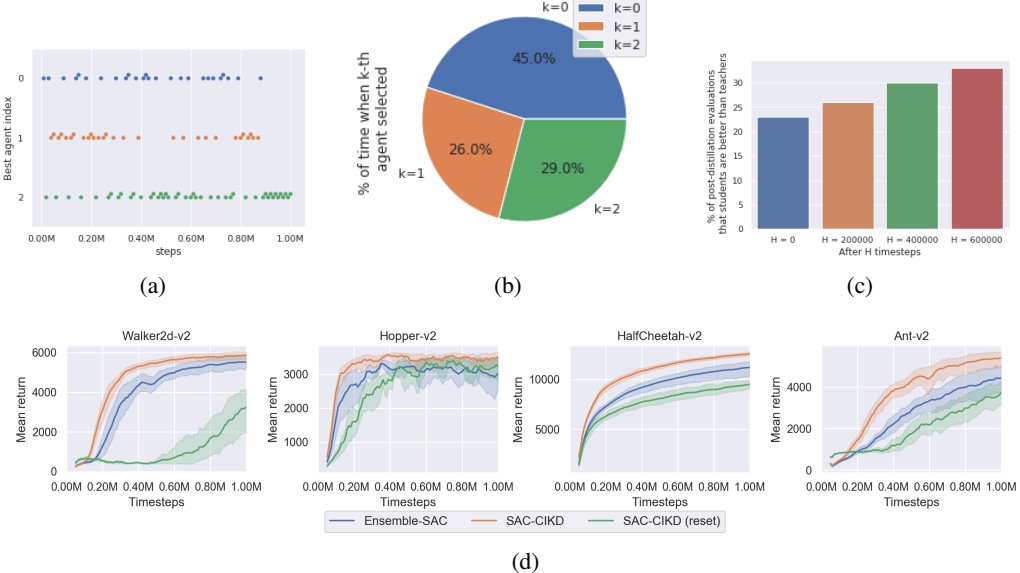

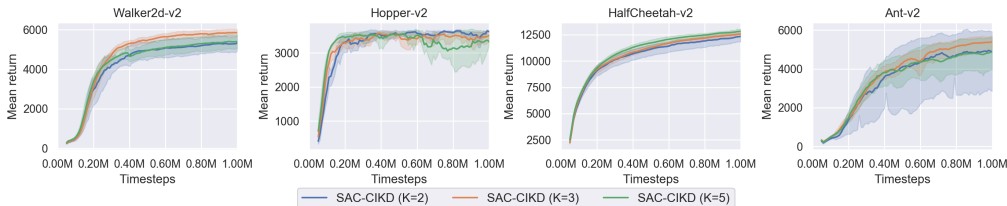

Figure 5: **(a) The distribution of selected teacher agent indices.** This figure shows that each agent has a chance to become the teacher. **(b) The proportion of time that each agent is selected to be the teacher over the entire training process.** This figure shows that there is no strictly dominant agent within the ensemble. **(c) The proportion of time that the students surpass the teachers after knowledge distillation.** This figure shows that as more experience is collected, inter-agent knowledge distillation enables students to become better than their teachers (see Section 4.4 for the detail of inter-knowledge distillation). **(d) The importance of students' inherent knowledge.** *SAC-CIKD (reset)* denotes the variant of our method where we randomly re-initialize students' parameters before knowledge distillation. This figure shows that students' inherent knowledge is crucial for the performance of our method.

## 5.6 ABLATION STUDY ON ENSEMBLE SIZE

In this section, we study the influence of the ensemble size $K$ as the ensemble size affects the scalability. We test our method with three ensemble sizes $K = 2$, $K = 3$, and $K = 5$. Figure 6 shows the performance of these configurations. We find that *SAC-CIKD* performs approximately the same across all three ensemble sizes. Even with an ensemble size of 2, we see better performance than *Ensemble-SAC* (as $K = 2$ is on-par with $K = 3$, which outperforms *Ensemble-SAC*, as shown before). Thus, our method can reap benefits even from small ensembles, and is not extremely sensitive to the ensemble size. However, investigating CIKD on a large scale ensemble is not the primary focus of this paper, and we leave its investigation for future work.

Figure 6: **Performance comparison under different ensemble sizes.** Three different ensemble configurations with 2, 3, and 5 agents lead to similar performance. This result shows that CIKD does not require a large ensemble size.

## 6 CONCLUSION

In this paper, we introduce collaborative inter-agent knowledge distillation (CIKD), a method that jointly trains an ensemble of RL agents while continually sharing information via knowledge distillation. Our experimental results demonstrate that CIKD improves the performance and data efficiency of a state-of-the-art RL method on several challenging MuJoCo tasks. Also, we show that knowledge distillation is more effective than the other approaches for knowledge sharing. We found that electing the best-performing agent to serve as the teacher plays a significant role in improving performance. Our investigation further showed that the combination of students' and teachers' knowledge is crucial for the performance. Finally, our ablation study showed that a large ensemble is not needed for improving performance.

CIKD open several avenues for future work. First encouraging diversity within the ensemble may lead to more efficient exploration [4, 25]. Additionally, while we used a simple uniform policy selection strategy, a more efficient policy selection strategy may further accelerate learning. Lastly, while our ensemble members used identical architectures, CIKD may benefit from using heterogeneous ensembles. For example, different networks may have different architectures that are conducive to learning different skills, which can then be distilled within the ensemble.

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

## A    DIFFERENT DISTILLATION SCHEMES FOR ACTOR-CRITIC METHODS

As actor-critic methods like SAC [19] train a critic in addition to the policy, three straightforward knowledge distillation schemes can be used: distilling both the policy and the critic, distilling the policy only, and distilling the critic only. In this section, we compare the both schemes to justify our choice of distilling both the policy and the critic. We plot the performance of both schemes in Figure 7, where *SAC-CIKD (policy only)* and *SAC-CIKD (critic)* stand for the variant of *SAC-CIKD* that distills the policy only and distills the critic only, respectively. We see that both *SAC-CIKD (policy)* and *SAC-CIKD (critic)* sufficiently improve the performance against the vanilla SAC while *SAC-CIKD* achieves the best performance. This result shows that distilling the critic and the policy simultaneously achieves superior performance.

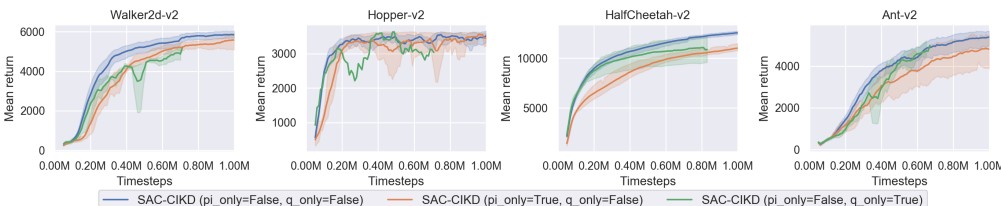

Figure 7: **Comparison of different distillation schemes.** *SAC-CIKD (policy only)* and *SAC-CIKD (critic)* denote the variants of *SAC-CIKD* that distill the policy only and distill the critic only, respectively. This figure shows that distilling the critic and the policy simultaneously is more effective.

### A.1    EXTENDED EXPERIMENTAL RESULTS

We run SAC-CIKD as well as each of our baseline algorithms for longer timesteps in this set of experiments. We ran for 1.5 Million timesteps, which is longer than our 1M timesteps in the body.

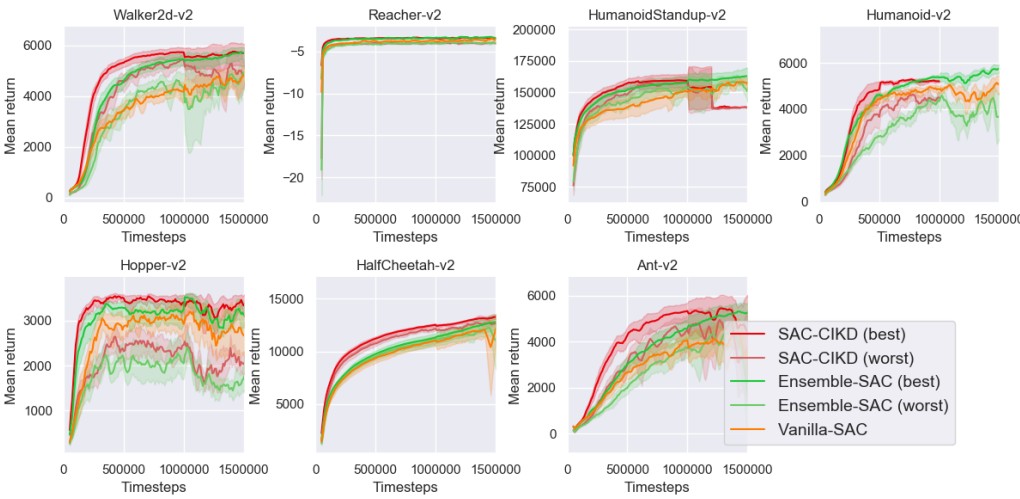

Figure 8: **Full performance evaluation of inter-agent knowledge distillation.** *SAC-CIKD* represents the implementation of our method upon SAC; *Vanilla-SAC* stands for the original SAC; *Ensemble-SAC* is an analogous variant of Osband et al. [37]'s method on *vanilla-SAC* (effectively *SAC-CIKD* without inter-agent knowledge distillation). See Section 5.2 for details.

### A.2    DOMINANCE IN THE ABSENCE OF KNOWLEDGE DISTILLATION

In this section, we extend our analysis in Section 5.5 with the goal to examine if there is a consistently dominated agent in *Ensemble-SAC* and *Vanilla-SAC* (see Section 5.2 for details). For *Ensemble-SAC*, we plot the index of the best-performing agent in the ensemble in Figure 9a and the propotions in Figure 9b. It can be seen that there is no consistently dominated agent in *Ensemble-SAC*, which is similar to the result of *SAC-CIKD* in Section 5.5. On the other hand, we plot the index of the best-performing agent, among three independently trained *Vanilla-SAC* agents (with different random

seeds), in Figure 9c. The proportions are presented in Figure 9d. It can be observed that there is a significantly dominated agent among these agents (i.e. $k = 2$).

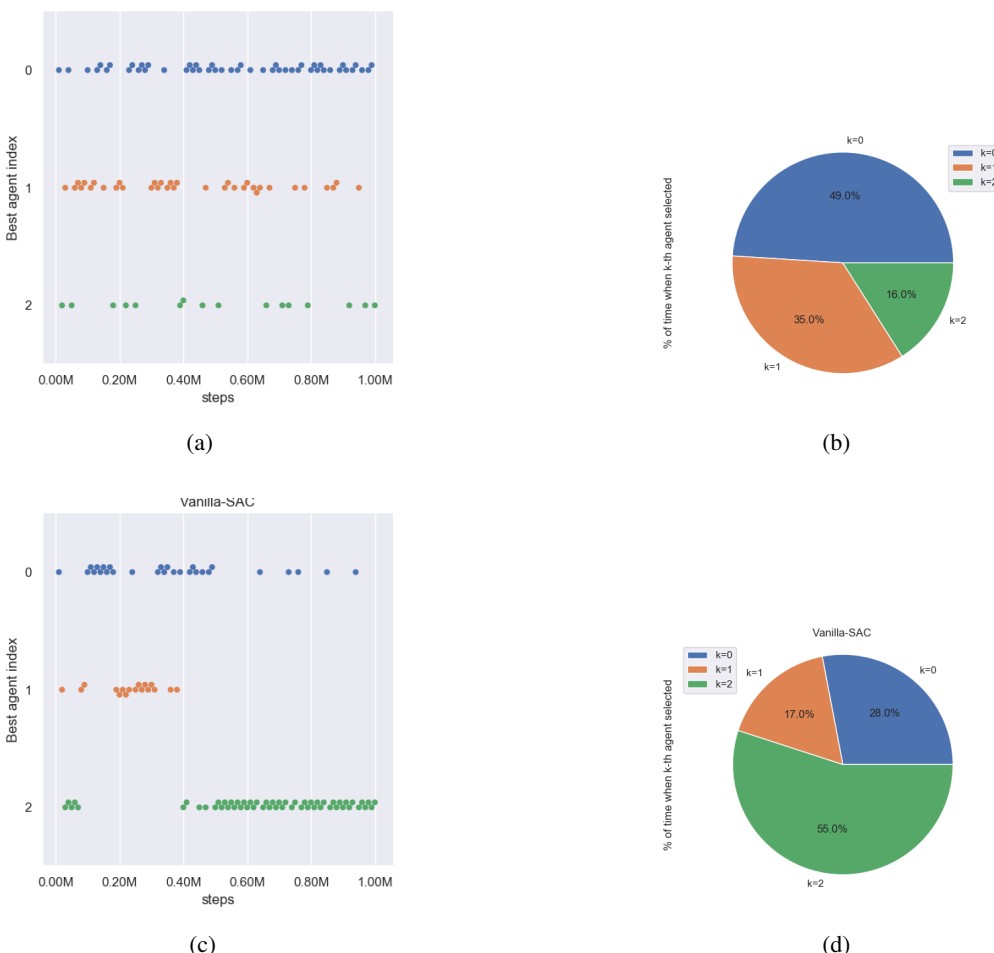

(a)

(b)

(c)

(d)

Figure 9: **(a) The distribution of selected teacher agent indices. (b) The proportion of time that each agent is selected to be the teacher over the entire training process. (c) The distribution of the best-performing agent among three agents *Vanilla-SAC* trained with different random seeds. (d) The proportion of time that each agent is the best-performing agent over the entire training process.**

