# OpenReview forum: "Collaborative Inter-agent Knowledge Distillation for Reinforcement Learning"
_ICLR.cc/2020/Conference — Reject_

### Official Review · AnonReviewer3 · 2019-10-09
**Official Blind Review #3**

**Rating:** 8

**Review:**

Summary:
This paper proposed an ensemble method (CIKD) that train multiple agents and
use knowledge distillation to transfer knowledge from the current best agent to
sub-optimal agents periodically.  According to the reported results, CIKD is a
simple yet effective approach to improve sample-efficiency and final performance.
The experimental results are sufficient, and the ablation studies are conducted thoroughly. It is shown that both selecting the best agent and using KD to
transfer knowledge are effective comparing to other naive alternatives.


I recommend the acceptance of this paper.

The paper proposed a novel approach (CIKD) to improve the sample-efficiency of the state-of-the-art. The proposed ensemble approach is aligned with our intuition, and it is effective. The authors proposed to train several agents at the same time and randomly select one of
the agents as a behavior policy during each rollout. Then the collected trajectory is used to update the policy of all agents. Meanwhile,
they keep tracking the performance of each agent and use the current best agent to conduct knowledge distillation to other agents periodically.

This paper first conducts experiments to show when consolidating
the SAC with CIKD, both of the final performance and sample-efficiency can be improved. Then a set of ablation studies verified the best agent selection strategy, and the knowledge distillation
strategy is necessary for the ensemble method.


Investigation on the reasons for improvement:
Though extensive ablation studies have shown the effectiveness
of each component of CIKD. It is still not clear why this approach
can be effective.
Intuitively, it is possible that the exploration from a set of agents would outperform
a single agent. The measure of exploration efficiency could help in explaining the results. Furthermore, better exploration not necessarily
leads to better performance and sample-efficiency. Does knowledge distillation serve as a better alternative to exploit existing data?

Model/algorithm agnostic
The proposed method is more convenient to be applied with off-policy approach when the policy is in the form of softmax. Is it also applicable
to other approaches?

Experiments:
How do you determine when to stop the KD process? As mentioned in section 5.5, if we conduce KD fully, all students would be just imitating
the teacher's behavior. It seems the key is to tune a good termination
threshold for each task? Are there any guidelines to set up this threshold?
Do you have some automatic way to terminate the KD procedure?


Minor:
L1, P5, "how to CIKD improves the sample efficiency"



**Experience Assessment:**

I have read many papers in this area.

**Review Assessment: Checking Correctness Of Derivations And Theory:**

N/A

**Review Assessment: Checking Correctness Of Experiments:**

I carefully checked the experiments.

**Review Assessment: Thoroughness In Paper Reading:**

I read the paper at least twice and used my best judgement in assessing the paper.

---

> ### Author Response · Authors · 2019-11-14
> **Response to Reviewer #3**
>
> We would like to thank the reviewer for reading our paper and providing feedback on our work. We will address the reviewer’s various points here.
>
> - “Though extensive ablation studies have shown the effectiveness of each component of CIKD. It is still not clear why this approach can be effective. Intuitively, it is possible that the exploration from a set of agents would outperform a single agent. The measure of exploration efficiency could help in explaining the results.”
>
> The purpose of the Ensemble-SAC baseline was to investigate how CIKD itself improves upon Ensemble-SAC, since Ensemble-SAC may naturally benefit from improved exploration upon a single SAC agent. In this way, we can decouple (to a certain degree), the benefits of an ensemble vs. the benefits of applying CIKD to an ensemble. In future work, it would be interesting to perform more experiments and analyses on the effect of improved exploration and the benefit of distillation (e.g., through plotting the state-visitation frequencies or distilling the knowledge from a separate hand-crafted dataset as opposed to agent data).
>
> - “Furthermore, better exploration not necessarily leads to better performance and sample-efficiency. Does knowledge distillation serve as a better alternative to exploit existing data?”
>
> It is unclear how to compare various approaches to exploiting existing data, since there is no general framework for data exploitation. However, we would like to highlight two of our experiments that investigated this question. Off-policy RL offers an obvious way to exploit experiences. In Section 5.3 (Fig. 3a), we performed an experiment where we tuned an Ensemble-SAC agent to perform additional off-policy RL updates. We found that using CIKD with Ensemble-SAC outperforms Ensemble-SAC with additional RL updates. Our second experiment, the “hard-copy” experiment (Fig. 3b, Section 5.3), copies the best teacher into the students rather than performing distillation. We found that distillation performs better than strictly hard-copying the best agent. Interesting directions for future work include performing additional analyses on various data exploitation methods.
>
> - “Model/algorithm agnostic: The proposed method is more convenient to be applied with off-policy approach when the policy is in the form of softmax. Is it also applicable to other approaches? “
>
> Our method is certainly applicable to other approaches. In particular, our KL Loss can be applied to other policy gradient approaches as long as the policy outputs a distribution and is differentiable, as is the case with most modern policy representations. In principle, CIKD can be applied to value-based approaches as well by changing the distillation loss from a KL-Loss to another loss, such as mean-squared-error (MSE). In fact, in our paper, we distill our critics using an MSE loss.
>
> - “How do you determine when to stop the KD process? As mentioned in section 5.5, if we conduce KD fully, all students would be just imitating the teacher's behavior. It seems the key is to tune a good termination threshold for each task? Are there any guidelines to set up this threshold? Do you have some automatic way to terminate the KD procedure?”
>
> We didn’t focus on optimizing the terminating threshold for distillation and found that CIKD worked quite well by randomly dividing the entire (bounded) replay buffer into several minibatches and performing distillation on all of these minibatches. If this process were to be repeated infinitely, this would amount to imitation learning. To verify that CIKD is not tantamount to pure imitation learning, we ran two key experiments. In one experiment (Section 5.5, Figure 5d), we tested an alteration of CIKD where we re-initialized the student networks prior to distillation. This amounts to pure imitation learning in that we have a randomly initialized student learning to directly imitate the teacher. We found that pure imitation learning fails to perform as well as CIKD. In Section 5.5 (Fig. 5c), we show that the student often outperforms the teacher after distillation. Note that outperforming the teacher is atypical in imitation learning, which further supports that CIKD does not reduce to imitation learning. Returning to the reviewer’s question, an interesting direction for future work is to investigate the tradeoff between pure imitation learning and a moderate amount of distillation. But in this work, we found that CIKD achieved good performance with straightforward distillation termination conditions and is in fact superior to distilling via pure imitation learning.
>
> - “Minor: L1, P5, ‘how to CIKD improves the sample efficiency’”
>
> We have corrected this mistake in the paper.

---

### Official Review · AnonReviewer2 · 2019-10-27
**Official Blind Review #2**

**Rating:** 6

**Review:**

This paper proposes an RL training procedure that maintains an ensemble of k policies and periodically pushes all the policies to be closer to the best performing one. The formulation, experiments and analysis are very clear and show a mild improvement over using the same underlying RL algorithm without the imitation part. The idea is close to many other proposed in the literature, but to my knowledge it is the first time this exact procedure is studied in detail.

The first piece of their approach is an off-policy RL algorithm. In their case, they use SAC. The second piece is adding an ensemble of policies (3 in their case), and randomly selecting one of them every time a rollout is collected, and using the collected rollout to update all the policies. This effectively implies 3 times more overall gradient updates compared to SAC. They call this ablation SAC-ensemble. Interestingly they only use the most recently collected trajectories to update all policies, and despite storing the rollouts in a replay buffer, they seem to only use the stored transitions for the imitation part described below. Some of their experimental results uses extra gradient steps, although it’s not clear if those gradient steps are also only on the last rollout collected, or on transitions sampled from the replay buffer as it is typical in off-policy RL methods. In general, I think the work could improve with more details about how much the policy training could improve by increasing the number of gradient steps on the full replay buffer.

The final piece of their method is selecting the best performing policy (or “teacher”) of the ensemble based on the recent experience, and update all other policies by executing some gradient steps on the KL divergence between them and the current “teacher”. They also try an experiment where the “teacher” is selected randomly, and it does surprisingly well in my opinion (specially realizing that the “HalfCheetah” experiments seem to not have all seeds run to convergence, please report the full results). I suspect that most of the benefit of their method comes from randomly perturbing the parameters of the policies in the ensemble. More thorough and careful experimentation needs to be carried out to investigate this direction. This is in fact not very surprising given the results of Evolutionary Strategy methods, or Population-based training (even if usually used for hyper-parameters adaptation).

Furthermore, the authors only run the environments for 1M steps, whereas in previous works some environments are shown to get higher return after more training steps. I would also encourage the authors to report the results in all the standard MuJoCo benchmarks for the ablations (even if it’s in the appendix) to better asses their claims.

Overall, this is a very well presented work, although it lacks some novelty and a few more thorough experiments to fully understand the improvements they show. I think this idea is worth sharing with the community, and I recommend a weak accept.

**Experience Assessment:**

I have published in this field for several years.

**Review Assessment: Checking Correctness Of Derivations And Theory:**

I carefully checked the derivations and theory.

**Review Assessment: Checking Correctness Of Experiments:**

I carefully checked the experiments.

**Review Assessment: Thoroughness In Paper Reading:**

I read the paper thoroughly.

---

> ### Author Response · Authors · 2019-11-14
> **Response to Reviewer #2**
>
> First, we would like to thank the reviewer for the time and effort given to the review, and for his/her valuable comments. We will address various points that the reviewer mentioned here.
>
> -“Interestingly they only use the most recently collected trajectories to update all policies, and despite storing the rollouts in a replay buffer, they seem to only use the stored transitions for the imitation part described below.”
> -“Some of their experimental results uses extra gradient steps, although it’s not clear if those gradient steps are also only on the last rollout collected, or on transitions sampled from the replay buffer as it is typical in off-policy RL methods”.
>
> In fact, we indeed do what the reviewer notes that we should do, i.e., we use “transitions sampled from the replay buffer as it is typical in off-policy RL methods”, for all of our experiments, as the original SAC algorithm does. Perhaps this misunderstanding stems from our pseudocode (Algorithm 1), which was written to be general. Though we allude to the use of experience replay for policy training in Section 4.2, we realize that the pseudocode is misleading to make one think that our policy is only trained by the most recent rollouts. We have rewritten our pseudocode to accurately reflect our experiments and the SAC algorithm.
>
>
> -“I suspect that most of the benefit of their method comes from randomly perturbing the parameters of the policies in the ensemble. More thorough and careful experimentation needs to be carried out to investigate this direction.”
>
> First, we would like to clarify that our results demonstrate that selecting the best agent to be the teacher has some benefit over choosing a random teacher. Perhaps we misunderstood, but we interpreted the reviewer’s mention of the “random perturbation” to mean the change in parameters after performing distillation with a random teacher.  We agree that this question is worth investigating, and we will address these in subsequent experiments in the future.
>
> Our hypothesis (which we have updated in the draft), outlined in Section 5.4, is that the reason that distillation from a random teacher performs quite well is because reducing the KL divergence between policies in the ensemble makes each policy better at learning from off-policy data generated from other members of the ensemble, by reducing the extrapolation error that comes from off-policy data distributions [11]. Thus, the distillation improves the quality of the off-policy RL updates, leading to better-than-expected performance for the agents.
>
> We intend to investigate this improvement by measuring the extrapolation error [11] (stemming from learning from off-policy data) before and after distillation to measure this effect. We can do so with the method used by Fujimoto et al. [11], where they measured the extrapolation error for DDPG, an off-policy actor-critic method applied to Mujoco tasks. We will also run additional ablations where we withhold some agents in the ensemble from distillation or distill from all members of the ensemble to a single agent.
>
> We would like to re-emphasize that these additional investigations are not fundamental to our core claims and results in the paper. These experiments are interesting supplementary experiments that better explain the reasons behind our performance improvements upon Ensemble-SAC. However, they will not change our core result which is that Ensemble-SAC augmented with CIKD gives improves performance across several Mujoco tasks.
>
> -“specially realizing that the “HalfCheetah” experiments seem to not have all seeds run to convergence, please report the full results”
> -“Furthermore, the authors only run the environments for 1M steps, whereas in previous works some environments are shown to get higher return after more training steps.”
>
> We intend to run experiments for longer training times and on all standard Mujoco tasks. Due to limited resources, we have prioritized (for the rebuttal) running experiments for longer training times. These experiments for longer training times are currently underway and we will post them to the rebuttal as soon as possible. We will also run experiments on more Mujoco tasks, though it is not likely that we will be able to complete them within the rebuttal period.

---

### Official Review · AnonReviewer4 · 2019-11-08
**Official Blind Review #4**

**Rating:** 3

**Review:**

This paper introduces a method for using an ensemble of deep reinforcement learning policies, where members of the ensemble are periodically updated to imitate the most promising member of the ensemble. Thus learning proceeds by performing off policy reinforcement learning updates for each individual policy, as well as some supervised learning for inter-policy imitation learning.

I start by what I view as the positive aspects about the paper:
1- The algorithm is quite simple (to understand and to implement).
2- Experimental results are performed on a variety of domains, and more importantly, each experiment is motivated by a question.

That said, I have some concerns about this paper which I list below:

1- Perhaps my biggest concern is that the approach is not motivated from a theory stand point. There has been interesting results in Osband's work [Osband, 2016] (and references therein) for randomized value functions which can serve as a foundation for this work. That said, a) Osband's results, at least immediately, are related to value-function based methods, as opposed to policy gradient b) the KL update which one could argue is the main and only significant contribution of the paper, is not justified by Osband or any other prior work c) there is not anything that this paper adds to the literature to better justify diversity through randomization and/or imitation learning based on the best member of the ensemble.

2- I have found various claims in the paper which are unclear, scientifically not true, or sometimes even contradicting. In Introduction, for example, the authors mention that the agent sometimes gets into a sub-optimal policy and may require a large number of interactions before escaping the sub optimal policy. How does gathering more data help to improve the policy? Either we are in a local maximum, which if we are doing gradient ascent, there is really not much we could do, or that we are in a saddle point, which we can escape by adding some noise to the gradient. [Jin,2017]

3- In section 4.3 the authors talk about on-policy methods requiring importance sampling (IS) ratios. To the best of my knowledge, IS is only used for off-policy learning. Can the authors provide a link to an on-policy method that does IS?

4- Again in section 4.3 authors claim and I quote "Using off-policy methods, all the policies in the ensemble can easily be updated, since off-policy update methods can perform updates from any \tau". But later on in Section 5.3 authors claim that "off-policy actor-critic methods (e.g. SAC) cannot fully utilize the other agent's or past experience." So which statement is true?

5- Again, the KL update is interesting, but is it even surprising that the KL update is necessary for an ensemble of policies updates using policy gradients? In the absence of this KL update, which the authors characterize as the method that Osband proposed, the policies could generally be arbitrarily far from one another. This means that each policy needs to perform policy evaluation using trajectories that are coming from other policies who in principle can be radically different than the policy we want to update. This means that updates will be quite "off-policy" which we know can really degrade the quality of the estimated gradient. This is perhaps why even choosing a random policy to update towards is providing "some" improvement. I think this is the real insight, but it is not really discussed at all in the paper.

6- On the same note, I do not think that one can say Osband's method is the same as CIKD but only without the KL update. Most notably, Osband's work was presented for value-function-based methods like DQN. These methods work fundamentally different than policy gradient methods, which rely on (near) on-policy updates to perform good policy improvements. In that sense, the presented results make sense, but I disagree with the framing of the results and how they are presented here.

7- In section 5.3, when the authors utilize more policy updates to have a fair comparison, are they retuning hyper parameters? Surely they need to do that, at least for hyper-parameters that are known to be super important such as the step size.

8- Overall I liked section 5.5 that is trying to dissect causes for improvement. However, it seems like that the "dominant agent" hypothesis has been rejected hastily, unless I misunderstood the experiment. The authors show that the notion of best is spread across different agents. But of course this will be the case in light of the KL update, since the policies are getting closer to one another. Can you redo the experiment in the absence of the KL update?

9- Have the authors thought about any connection between this and genetic algorithms? In genetic algorithms, the idea is the next set of candidates are chosen based on the most promising candidates in the current iteration. CIKD seems like a soft implementation of this idea.

In light of the comments above, I am voting for weak rejection, though as I said before, I do see some interesting things in this paper. I encourage the authors to think about CIKD from a theoretical lens in the future.

**Experience Assessment:**

I have read many papers in this area.

**Review Assessment: Checking Correctness Of Derivations And Theory:**

N/A

**Review Assessment: Checking Correctness Of Experiments:**

I assessed the sensibility of the experiments.

**Review Assessment: Thoroughness In Paper Reading:**

I read the paper at least twice and used my best judgement in assessing the paper.

---

> ### Author Response · Authors · 2019-11-09
> **Response to Reviewer #4 (1/2)**
>
> Question/Comment #1 Response:
>
> First, we would like to thank the reviewer for his/her detailed/thorough review of our paper. Regarding the biggest concern being that the paper is not motivated from a theory standpoint, it is unclear whether the reviewer is suggesting that we should have theoretical results or whether we should have theory motivating our method. We certainly think that rigorous, empirical contributions are extremely valuable contributions to the field, and there have been several empirical papers published at ICLR.
>
> Regarding 1a), it is true that our empirical results are on soft actor-critic, an actor-critic method, whereas Osband’s results are on value-function based methods. Is there a concern to be raised in 1a that we may address?
>
> Regarding 1b), we respectfully disagree that the KL distillation update is the only significant contribution of the paper. Our contribution is an empirical demonstration that combining the training of an ensemble of RL agents with periodic distillation between the members of the ensemble can significantly improve performance, and it is backed by several experiments. However, it is true that the distillation itself is primarily carried out through a combined KL update for actors and mean-squared error for the critics. However, we disagree with the characterization that the loss functions alone are the main/significant contribution, as it diminishes the importance of executing this gradient update in the context of training an ensemble with periodic distillation, which to our knowledge nobody has attempted. The point 1c seems to implicitly suggest (please correct us if we are wrong) that our paper’s aim is to justify diversity through randomization and/or imitation learning. Our paper builds off of ensemble RL, and our paper’s aim is to provide a method that improves upon a vanilla/standard form of ensemble RL. So our experimental results are not meant to further highlight the benefit of diversity beyond existing literature that justifies it. If we overemphasized the importance of diversity to the point where it mischaracterizes our contribution, we apologize. Our goal in motivating diversity was because we are considering settings where we are training an ensemble of RL agents. That is, our paper first motivates diversity since diversity is a precursor to the application of our actual method/contribution.
>
> Question/Comment #2 Response:
>
> We apologize if we are unclear, and are happy to address any individual instances of unclear wording that the reviewer presents. In RL, exploration typically refers to trying actions randomly or randomly perturbing policy parameters to collect trajectories in the environment. To address the reviewer’s question, gathering more data through exploration can help improve the policy by adding some noise to the gradient, which is similar to the reviewer’s note. Our point was that for the agent to improve, it needs to be rewarded for “good” behavior. An RL agent typically explores policies that are close to its greedy policy (e.g., sampling from a gaussian policy or adding the noise to the greedy actions). If that greedy policy is poor, it can require quite a bit of exploration in order to acquire those good experiences. We understand the reviewer’s concern here, and have reworded those sentences in the paper.
>
> Question/Comment #3 Response:
>
> Thank you for pointing this out. This is correct. It was our intention to suggest that we can take traditional on-policy updates and have them use off-policy data for learning by applying importance sampling. However, these are then off-policy algorithms, not on-policy algorithms. We have removed this from the paper.
>
> Question/Comment #4 Response:
>
> In Section 4.3, we are speaking theoretically, whereas in Section 5.3, we are speaking empirically. That is, theoretically, off-policy methods can update their policies by experience generated by any policy (e.g., human experts, past experience, and the other agents’ policies). However, in Section 5.3, we are saying that SAC, in practice, cannot fully benefit from the past experience (similar to the reviewer’s comments in point 5). Recent work [11] which we cite in our paper, supports this claim, demonstrating that DDPG, an off-policy critic method failed to learn well from data that deviates too much from the agent’s current policy. We have updated the draft based on these comments.

---

> > ### Author Response · Authors · 2019-11-09
> > **Response to Reviewer #4 (2/2)**
> >
> > Question/Comment #5 Response:
> >
> > Whether or not it is surprising that the KL update is necessary is quite subjective. However, given that this question (i.e. Point 5) is listed under the reviewer’s concerns about the paper, we would urge the reviewer to consider our contribution. When we have an empirical hypothesis, and we test it, and demonstrate that it is useful for improving performance, that is valuable to the community. We do not think that a hypothesis needs to be surprising for it to be a valuable contribution to the field (if that was a concern).
> >
> > Regarding the reviewer’s point about why selecting a random teacher provides some improvement, we agree with the reviewer, and appreciate the comment. We viewed our random teacher experiment as an auxiliary experiment to our core result of using the best teacher, and thus did not devote as much text to discussing this experiment. However, the reviewer’s comments are interesting and important, and we have updated the paper to reflect those comments.
> >
> > Question/Comment #6 Response:
> >
> > The core idea of Osband’s method is to combine several value functions, which each induce a policy, and have these individual policies act in the environment and generate trajectories, which are then used to train all the value functions off-policy.  Our Ensemble-SAC similarly consists of several agents/policies, which act in the environment and generate trajectories, which are then used to train all the agents off-policy.
> >
> > We did not presume to say that Osband’s method is the same as CIKD but without the KL update. Our wording was that it was “effectively equivalent to CIKD-RL without inter-agent knowledge distillation”, which we realize is strong wording. We have rephrased this in the paper to indicate that Ensemble-SAC is the natural analog to Osband’s method in this setting.
> >
> > Question/Comment #7 Response:
> >
> > Actually, we did hyperparameter searches (on learning rate) for Ensemble-SAC (extra) and Vanilla-SAC (extra) and found out that the default hyperparameters are the best. Presumably, a smaller learning rate should be used with extra policy updates. However, it turned out that neither a smaller nor larger learning rate are better than the default hyperparameters reported in the original SAC paper.
> >
> > Question/Comment #8 Response:
> >
> > The dominant agent experiment is meant to test whether a single agent is consistently better than other agents in the ensemble. For us it was not obvious that the KL update will necessarily cause the best agent to change so frequently within the ensemble. While the KL update brings the policies closer, it certainly doesn’t suggest that one agent should surpass another after a KL update. In particular, the KL update is unidirectional, so if an agent A is better than agent B, then we perform updates on agent B. While we expect agent B’s policy to grow more similar to A’s policy, we wouldn’t necessarily expect it to surpass A, especially considering that B is worse than A before the KL update. It would be useful to run an additional experiment in the absence of a KL update to see whether a single agent is consistently dominant. We will try to get these results before the end of the rebuttal period.
> >
> > Question/Comment #9 Response:
> >
> > In an abstract sense, we are related to genetic algorithms in that we share knowledge between individuals, in our case an ensemble of RL agents, and in the case of genetic algorithms, a population of genotypes (candidate parameters to a certain problem in genetic algorithms). However, there are several differences in the details between CIKD and genetic algorithms. First, we use knowledge distillation for sharing knowledge, while typical genetic algorithms use crossover, which usually randomly exchanges the elements between two sequences.
> >
> > Secondly, we use RL and distillation to optimize each individual while typical genetic algorithms solely use mutation and crossover (i.e., randomly exchanging the elements between sequences). While distillation is somewhat related to crossover, typically crossover is a destructive process, either explicitly replacing the structure or parameters of the individual, unlike distillation, whose parameter changes are through gradient updates. Note that, as we showed in Figure 5d, the destruction of parameters before distillation harms the performance, which suggests the advantage of being able to preserve the learned knowledge.
> >
> >
> > Again, we would like to thank the reviewer again for providing valuable feedback and comments which we used to improve the paper.

---

> > > ### Author Response · Authors · 2019-11-15
> > > **[Update] Response to Reviewer #4**
> > >
> > > [Update] Question/Comment #8 Response:
> > > We have attached the new experimental results in the appendix.

---

### Public Comment · ~Kai_Li2 · 2019-09-28
**How**

Very interesting idea, how is this method compared to <Population Based Training of Neural Networks>

---

> ### Author Response · Authors · 2019-11-04
> **Compare**
>
> Thank you for your interest in our work. Population-Based Training of Neural Networks (PBT) is similar to our work at a high level in that it similarly employs multiple agents for training.
>
> We appreciate you for mentioning a related work. Population-Based Training of Neural Networks (PBT) is similar to our work in an abstract sense. PBT similarly employs multiple agents for training. However, our work differs from PBT in multiple ways.
>
> First, the goal of PBT is to optimize the hyperparameters online. However, our work aims to optimize an ensemble of policies given the same hyperparameters. Thus, PBT can be incorporated into CIKD, optimizing the hyperparameters of CIKD.
>
> Secondly, the core idea is different despite the similarity at an abstract level. Resembling evolutionary algorithms, PBT searches for the optimal set of hyperparameters via mutation, selection, and reproduction. Each set of hyperparameters is considered as an individual in the population. PBT iteratively mutates (i.e. randomly perturbs) the existing hyperparameters, then selects a group of top-ranked agents, and finally reproduces the population using the selected agents. Differing from PBT, our work does not require mutation and reproduction. We instead focus on improving the existing agents via distilling the knowledge of the selected best-performing agent. For a more detailed comparison, the reader can refer to Section 2 in our paper.

---

### Author Response · Authors · 2019-11-15
**Status update for paper**

We would like to thank all the reviewers for their helpful comments. We have provided responses to all of the reviewers, and have updated our paper in response to the reviews. In particular, we have improve the presentation of the material, and have run additional experiments. These additional experiments have been added to the Appendix, Section A1. Unfortunately, we were unable to complete all of the reviewers’ experiments, and were only able to run SAC-CIKD for 1.5M timesteps on the additional domains, as opposed to 3M, despite the fact that our GPUs have been continuously running. These partial results are in Appendix A1. We will expand our set of experiments in the future.

---

### Decision · Program_Chairs · 2019-12-19

**Decision:**

Reject

**Comment:**

The paper introduces an ensemble of RL agents that share knowledge amongst themselves. Because there are no theoretical results, the experiments have to carry the paper.  The reviewers had rather different views on the significance of these experiments and whether they are sufficient to convincingly validate the learning framework introduced. Overall, because of the high bar for ICLR acceptance, this paper falls just below the threshold.